# How Spatial Resolution of Remote Sensing Image Affects Earthquake Triggered Landslide Detection: An Example from 2022 Luding Earthquake, Sichuan, China

Yu Huang [1,2], Jianqiang Zhang [2,*], Lili Zhang [2,3], Zaiyang Ming [1,2], Haiqing He [1], Rong Chen [2], Yonggang Ge [2] and Rongkun Liu [4]

1   School of Geomatics, East China University of Technology, Nanchang 330013, China
2   Key Laboratory of Mountain Hazards and Earth Surface Process, Institute of Mountain Hazards and Environment, Chinese Academy of Sciences, Chengdu 610041, China
3   University of Chinese Academy of Sciences, Beijing 100049, China
4   School of Environment and Natural Resources, The Ohio State University, Columbus, OH 43210, USA
*   Correspondence: zhangjq@imde.ac.cn

**Abstract:** The magnitude 6.8 Luding earthquake that occurred on 5 September 2022, triggered multiple large-scale landslides and caused a heavy loss of life and property. The investigation of earthquake-triggered landslides (ETLs) facilitates earthquake disaster assessments, rescue, reconstruction, and other post-disaster recovery efforts. Therefore, it is important to obtain landslide inventories in a timely manner. At present, landslide detection is mainly conducted manually, which is time-consuming and laborious, while a machine-assisted approach helps improve the efficiency and accuracy of landslide detection. This study uses a fully convolutional neural network algorithm with the Adam optimizer to automatically interpret the aerial and satellite data of landslides. However, due to the different resolutions of the remote sensing images, the detected landslides vary in boundary and quantity. In this study, we conducted an assessment in the study area of Wandong village in the earthquake-affected area of Luding. UAV images, GF-6 satellite images, and Landsat 8 satellite images, with a resolution of 0.2 m, 2 m, and 15 m, respectively, were selected to detect ETLs. Then, the accuracy of the results was compared and verified with visual detection results and field survey data. The study indicates that as the resolution decreases, the accuracy of landslide detection also decreases. The overall landslide area detection rate of UAV imagery can reach 82.17%, while that of GF-6 and Landsat 8 imagery is only 52.26% and 48.71%. The landslide quantity detection rate of UAV imagery can reach 99.07%, while that of GF-6 and Landsat 8 images is only 48.71% and 61.05%. In addition, for each landslide detected, little difference is found in large-scale landslides, and it becomes more difficult to correctly detect small-scale landslides as the resolution decreases. For example, landslides under 100 $m^2$ could not be detected from a Landsat 8 satellite image.

**Keywords:** landslide detection; remote sensing images of different resolutions; fully convolutional neural network; Adam optimizer; Luding earthquake

## 1. Introduction

Landslides are the most common geological hazards in mountainous areas, which are characterized by large numbers, wide distribution, and severe destruction. The factors that induce landslides include rainfall, earthquake, snow melt, human activities, etc. Among them, landslides that are induced by strong earthquakes are higher in number, distribution range, and scale than any other types of landslides, and they have the characteristics of a concentrated and continuous distribution. Earthquake-triggered landslides (ETLs) often cause great damages to roads, houses, farmlands, oil and gas pipelines, and water management facilities in the earthquake area, resulting in large casualties and seriously affecting the post-earthquake rescue and post-disaster reconstruction. For example, the

2008 Wenchuan Ms 8.0 earthquake in Sichuan triggered more than 15,000 geological hazards [1], which caused more than 20,000 deaths. Landslides accounted for up to 40% of all potential hazard sites induced by the earthquake. At some point, the extent and impact of earthquake-induced geological hazards exceed the hazards that are directly caused by the earthquake [2].

The investigation of ETLs is one of the most urgent tasks for post-earthquake emergency relief and subsequent post-disaster reconstruction. Visual detection is the mostly used method to catalog ETLs, which is time-consuming and laborious, and often cannot meet the needs of earthquake emergency relief. Therefore, there is an urgent demand for automatic interpretation for investigating ETLs.

For example, Wen et al. used a Landsat series remote sensing data and digital elevation data (DEM) to conduct an in-depth analysis of the feature characteristics of landslide areas such as Mao County in Sichuan and Nayong in Guizhou [3]. Given the fact that the damage to topography and landforms by landslide disasters shows a similar pattern in remote sensing data, all the landslides had been detected by analyzing such a pattern [4–6]. Ramdhoni et al. performed a landslide extraction by the Smorph method using a slope and slope shape to establish the corresponding transformation matrix, and the landslide extraction accuracy reached 79.54% [7]. In the field of landslide detection automation, machine learning techniques such as support vector machines, artificial neural networks, and deep learning have been widely used. Sameen et al. proposed a method for landslide detection that used residual networks to enhance the same designed network through fusing feature information to obtain better performance [8]. Long et al. used the Mianyuan River basin, the extreme seismic area of the Wenchuan earthquake, as the study area to automatically detect geological hazards' information in the basin. They used the maximum likelihood method and random forest algorithm based on high-resolution multitemporal satellite images such as RapidEye to quantitatively analyze the spatial and temporal evolution trends of geological hazards after the earthquake [9]. Ye et al. used hyperspectral remote sensing data to detect landslides based on a deep learning framework with constraints (DLWC) [10]. Their method improved the detection results through logistic regression (LR) by combining the detected image features with additional constraints as input. In recent years, in the field of machine learning on ETL detection, the neural network approach is more common [11–16]. The advanced development of deep learning techniques such as the convolutional neural network (CNN) has been widely successful in extracting information from images and it surpasses the other traditional methods [17]. Liu et al. used a U-Net neural network and combined it with a GEE large remote sensing platform to achieve the rapid detection of large-area co-seismic landslides with a detection accuracy of about 70% [18]. Most of the articles used medium-resolution images, and since ETLs often cause severe surface damage and land cover changes, all the above studies have shown the feasibility of remote sensing automatic detection technology in ETL investigation. However, an accurate comparison of multiple types of image effects with machine learning models is lacking.

With the introduction of different discriminative models, despite the fact that landslide automatic detection technology has been developed, there are still some problems, especially that most of the current studies are based on a single kind of remote sensing images for detection research. For ETLs, due to their wide distribution, a large number of remote sensing images of different types and resolutions are often used in emergency investigations [19–21]. Additionally, such differences in landslide detection from multisource remote sensing images are rarely addressed.

On 5 September 2022, at 1252 h, an MS6.8 magnitude earthquake occurred in Luding County, Ganzi Prefecture, Sichuan Province, China, with a depth of 16 km. The epicenter was 29.59° N, 102.08° E. The seismic source mechanism of the earthquake is resolved as a walking-slip type rupture. As of 17:00 on 11 September 2022, a total of 93 people were killed by the earthquake. The earthquake-affected area is located in the Hengduan Mountains on the southeastern edge of the Tibet Plateau, which is a typical alpine valley

area characterized by short and steep slopes, high altitudes, and broken rocks. These geomorphological features caused a large number of landslides after the earthquake. In this paper, the Wandong village in the Luding earthquake area was selected as the study area. Remote sensing images with different resolutions, such as UAV images, GF-6 satellite images, and Landsat 8 satellite images were used to automatically detect landslides using the full convolutional neural network (FCN). The differences in the automatic ETL detection accuracy from different images were quantitatively compared, and the reasons for these accuracy differences were analyzed. This study provides support in terms of the applicability and effectiveness of deep learning methods for further improvement in the field of ETL detection. It also holds a reference value for assessing the integrity of acquired landslide catalogs based on remote sensing images with different resolutions.

## 2. Study Area

### 2.1. Overview of the Study Area

The study area of this paper is located in the village of Wandong, Detuo Town, Luding County. It is 9.43 km from the earthquake center and within the earthquake intensity zone of a magnitude of eight. The village is on the east side of the seismogenic fault Xianshui River Rift, between the Xianshui River Rift and the Jinping Mountain Rift Zone where the Dadu River flows through its eastern part (Figure 1). The study area covers 102.05°~102.26° E and 29.33°~29.70° N, with an area of about 9.26 square kilometers. The maximum elevation of the study area is 2767 m, the minimum elevation is 978 m, and the maximum difference in elevation reaches 907 m. The slopes in the area are steep, with 8% of the slopes less than 15°, 17% of the slopes between 15° and 25°, 40% of the slopes between 25° and 35°, 22% of the slopes between 35° and 45°, and 13% of the slopes greater than 45°.

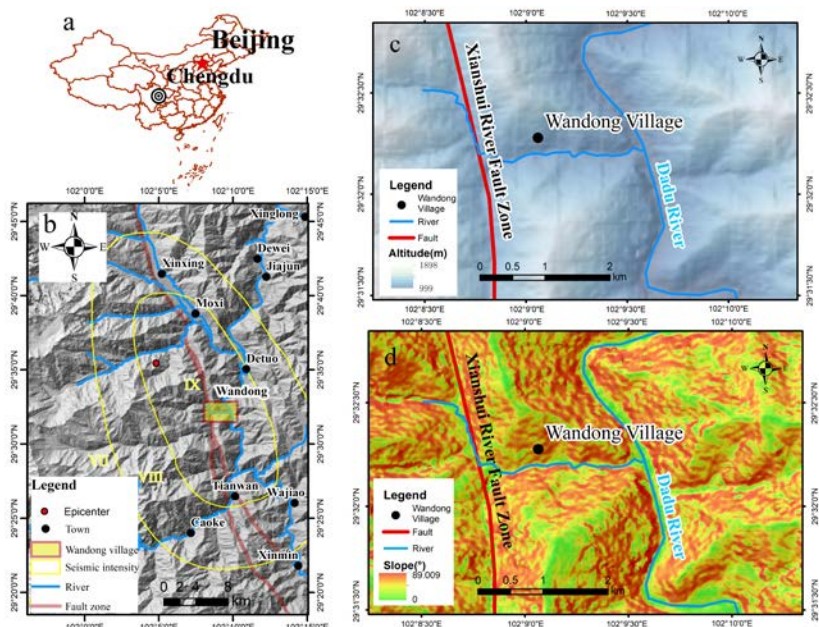

**Figure 1.** Schematic map of the disaster area (**a**). location of study area (**b**). hill shade map of study area (**c**) topographical map of study area. (**d**). slope map of study area.

After the Luding earthquake, the Institute of Mountain Hazards and Environment, Chinese Academy of Sciences used the visual detection of UAV images to investigate the ETLs in the region, which was further verified by visual investigation, and acquired a total of 434 ETLs with a landslide area of 1.49 km².

*2.2. Multisource Remote Sensing Data*

To compare the differences between images of different resolutions for ETL detection, three groups of images with different resolutions were sourced, namely, the UAV image, GF-6 image, and Landsat 8 image (Table 1). The UAV image has the highest resolution of 0.2 m, which was taken on 6 September 2022. The UAV image is a set of stitched and calibrated orthophotos of drone images provided by Sichuan Geographic Information Bureau. The GF-6 image with a resolution of 2 m was taken on 10 September 2022 while the Landsat 8s 15 m resolution image was taken on 11 September 2022. The Landsat image used in this paper is a fusion of RGB and 15 m panchromatic bands. Different features of the landslides are shown in the images with different resolutions (Figure 2). In the UAV image, the landslide boundary is clear, and the granular texture formed by broken rocks of the landslide body can be detected; in the GF-6 image, the landslide boundary is clear, but the landslide body does not show textural features; while in the Landsat 8 image, the boundary of the landslide body is more blurred. Smaller landslides (Figure 3) are still clearly identifiable in the UAV image, while they are fuzzier in the GF-6 image and cannot be shown in the Landsat 8 image.

**Table 1.** Basic parameters of remote sensing image.

|  | UAV | GF-6 | Landsat 8 |
|---|---|---|---|
| Resolution (m) | 0.2 | 2 | 15 |
| Acquisition time | 6 September 2022 | 10 September 2022 | 11 September 2022 |
| Type | Photos | RGB true color composite image | Multispectral image |
| Source | Sichuan Geographic Information Bureau | Aerospace Information Research Institute, Chinese Academy of Sciences | NASA |

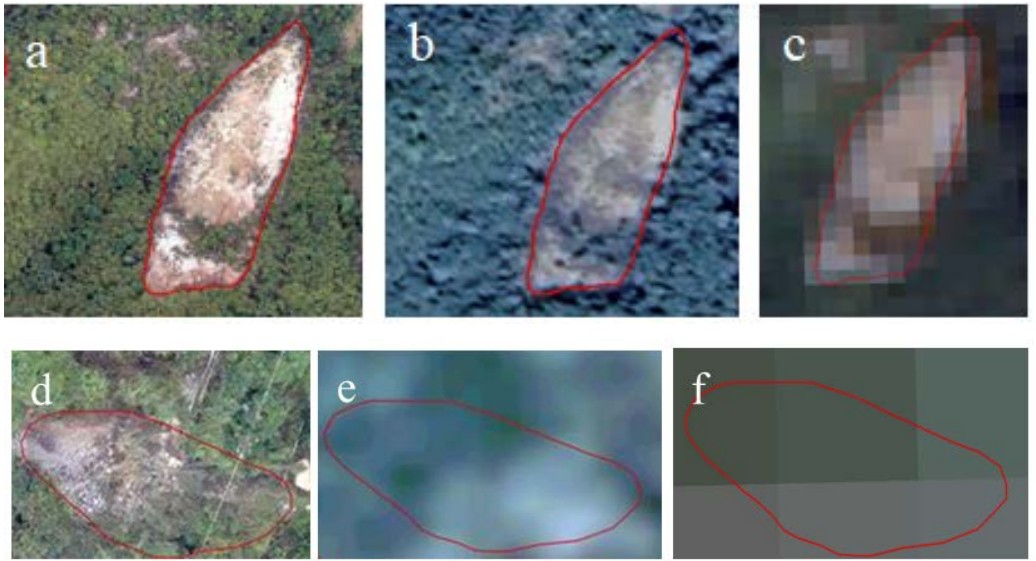

**Figure 2.** Schematic diagram of different scales landslides in each image (Landslide 1, covering an area of 21,680 m$^2$, located at 102°9′6″E, 29°31′53″ N; (**a**) clear landslide in UAV image (**b**); clear landslide in GF-6 image (**c**); clear landslide in Landsat 8 image; Landslide 2, with an area of 419 m$^2$, location: 102°8′27″ E, 29°31′54″ N (**d**); the landslide is relatively identifiable in the UAV image (**e**); the landslide is fairly vague in the GF-6 image (**f**); the landslide cannot be displayed in the Landsat 8 image).

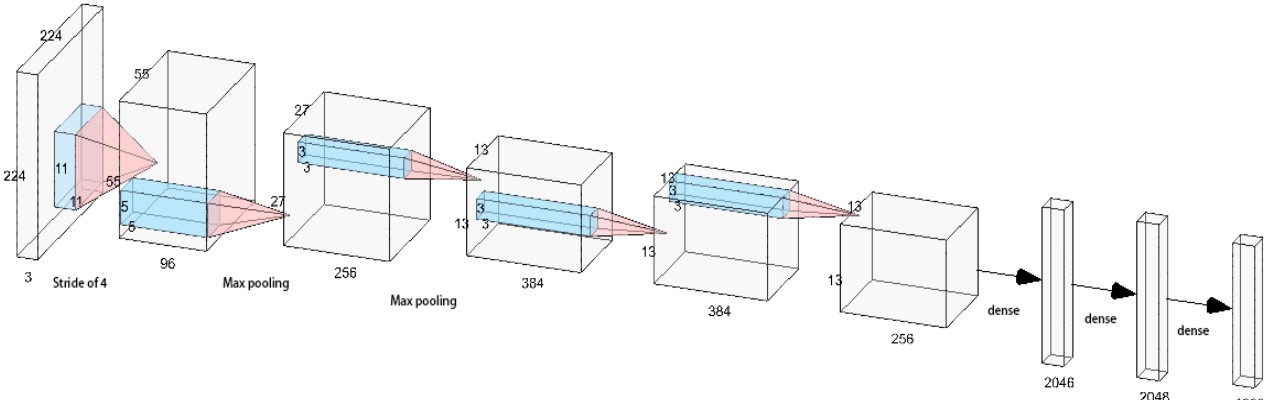

**Figure 3.** Schematic diagram of FCN.

### 3. Methods

#### 3.1. Deep Fully Convolutional Neural Networks

A fully convolutional neural network (FCN) was used for ETL detection in this study. The approach's main function is to obtain the function mapping between input and output variables, which are all quite tedious, complex, and non-linear. By constantly adjusting the number of neurons in the input, output and hidden layers of the network and the connection weights between the layers in the network, the requested non-linear function mapping relationship can be converged. After several adjustments of the network structure and experimental validation, the model structure shown in Figure 3 was designed by considering the training accuracy and learning efficiency. In this paper, a total of four convolutional layers were set up and the parameters were updated using the Adam optimizer.

#### 3.2. Adam (Adaptive Moment Estimation) Optimizer

For training time and validation score by the traditional iterative gradient descent algorithm, either random gradient descent or batch gradient descent algorithms have disadvantages. The batch gradient descent method is slow in training when the sample size is large; the random gradient descent method uses only one sample to decide the gradient direction, which leads to a solution that might not be the optimal. Therefore, this paper used Adam's algorithm instead of the traditional iterative gradient descent algorithm for landslide detection.

The Adam algorithm is an optimization of the gradient descent Adagrad algorithm [22] and the Rmsprop algorithm [23]. Additionally, the Adam algorithm has the advantages of not being affected by the gradient scaling transformation when the parameters are updated; it can automatically adjust the learning rate, be easily implemented, and has efficient computation rate [24]. Therefore, the Adam algorithm was chosen in this paper as the optimizer to improve the accuracy of the algorithm. The specific steps of Adam are as follows:

1.  Calculate the $J$ function derivative at moment $t$, i.e., the gradient:

$$g_t = \nabla_\theta J(\theta_{t-1}), \tag{1}$$

2.  Using the idea of momentum algorithm, the gradient of the current step is calculated by considering the historical momentum and the current gradient together:

$$m_t = beta_1 \times m_{t-1} + (1 - beta_1) \times g_t, \tag{2}$$

where $m_t$ represents the first-order exponential smoothing value of the historical gradient, and the $beta_1$ coefficient is the exponential decay rate, with a default value of 0.9.

3. Referring to the previous Rmsprop algorithm, the first-order exponential smoothing value of the gradient squared is calculated. This step calculates the variance of the gradient:

$$v_t = beta_2 \times v_{t-1} + (1 - beta_2) \times g_t\text{^}2, \qquad (3)$$

where $v_t$ represents the first-order exponential smoothing value of the gradient square, The $beta_2$ coefficient is the exponential decay rate. The weighted mean of the gradient square is 0.999 by default.

4. The gradient parameters are updated as follows:

$$\theta\_t = \theta\_(t-1) - a \times m\text{^}\_t / \left( \sqrt{(v\text{^}\_t)} + \varepsilon \right) \qquad (4)$$

where $a$ is the initial learning rate of 0.001, $\varepsilon$ is a constant $10^{-8}$ to avoid a divisor of 0. The update of gradient can be adaptively adjusted from the two angles: gradient mean and gradient squared, which also play a certain role of annealing.

5. At the initial stage of training, since $m_0$, $v_0$ is initialized to 0, which will cause $m_t$ to be biased towards 0; therefore, it is necessary to correct the bias $m_t$, $v_t$, to solve the problem at the initial stage of training. The formula is as follows:

$$\hat{m}_t = m_t / \left( 1 - beta_1^t \right), \qquad (5)$$

$$\hat{v}_t = v_t / \left( 1 - beta_2^t \right), \qquad (6)$$

In this study, the images of the study region were being matched and segmented, and the study region was divided into $500 \times 500$ pixel point images to produce accurate landslide inventory images, the $500 \times 500$ pixel range with an actual area of 10,000 m$^2$ allows for more efficient image segmentation while providing sufficient detail. To ensure the same feature details in divided areas, before segmentation, we resampled GF-6 and Landsat 8 image to the same size as the UAV image. The landslide results of all inventory images are set as raster to correspond to the images. Then, 70% of the inventory images were randomly selected as the training set for training, and the tested results were then stitched and calibrated to get the final overall landslide detection area. All workflows are implemented in python.

## 4. Comparison of Landslide Area Detection

### 4.1. Confusion Matrix

The deep fully convolutional neural network model is used to detect landslides from UAV images, GF-6 images, and Landsat 8 images, respectively. The UAV images detected 1.43 km$^2$ of landslides, which was the closest to the actual 1.49 km$^2$, while the GF-6 and Landsat 8 images deciphered a relatively smaller landslide area of 1.12 km$^2$ and 1.32 km$^2$. (Figure 4)

In this study, a confusion matrix was constructed for detecting landslides and non-landslides from different types of remote sensing images (Table 2). In the matrix, P represents positive areas, i.e., the areas where landslides occurred, and N represents negative areas, i.e., the areas where no landslides occurred. TP represents the true positive areas, i.e., the areas where both actual and predicted cases are positive, which means the correctly interpreted landslide areas; TN represents true negative areas, i.e., the areas when both the actual and predicted cases are negative, meaning the correctly interpreted non-landslide areas; FN represents pseudo-negative areas, i.e., the areas where the actual condition is positive but the predicted condition is negative, showing the incorrectly interpreted landslide areas; and FP represents pseudo-positive areas, i.e., the areas where the actual condition is negative but the predicted condition is positive, showing the incorrectly interpreted non-landslide areas.

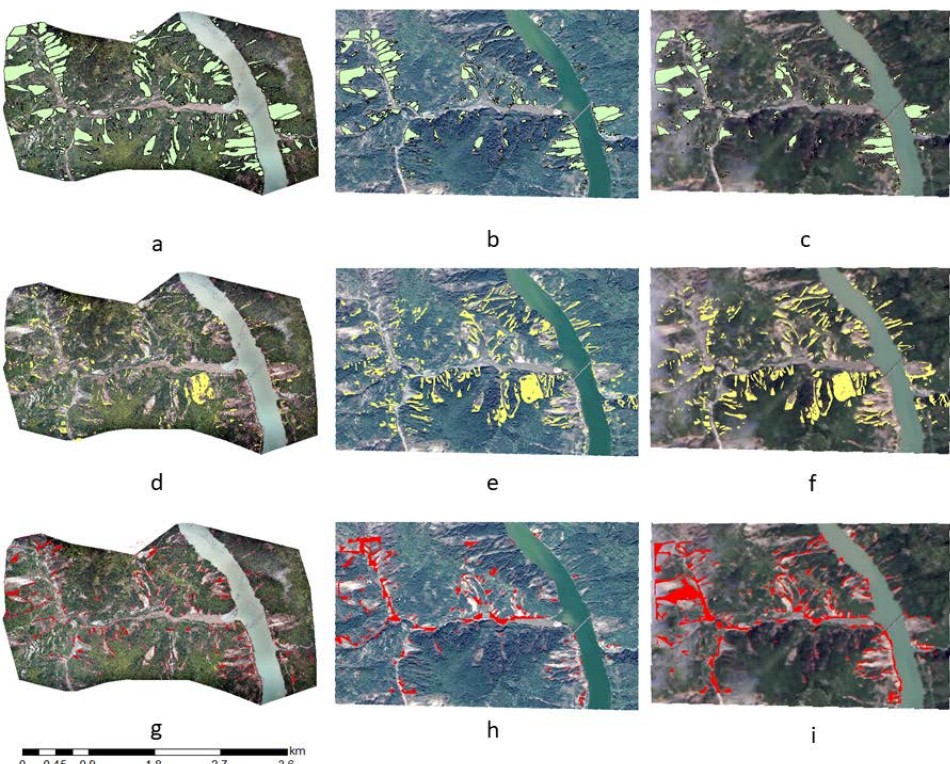

**Figure 4.** Detected areas in each image (((**a**) correctly detected areas in UAV; (**b**) correctly detected areas in GF-6; (**c**) correctly detected areas in Landsat 8; (**d**) undetected areas in UAV; (**e**) undetected areas in GF-6; (**f**) undetected areas in Landsat 8; (**g**) misidentified areas in UAV; (**h**) misidentified areas in GF-6; and (**i**) misidentified areas in Landsat 8).

**Table 2.** Landslide detection area results from remote sensing images.

| | | Landslide (Detection) (km$^2$) | | Non-Landslide (Detection) (km$^2$) | |
|---|---|---|---|---|---|
| Landslide (1.43 km$^2$) | UAV | TP | 1.23 | FN | 0.26 |
| | GF-6 | | 0.79 | | 0.70 |
| | Landsat 8 | | 0.73 | | 0.76 |
| Non-landslide (7.77 km$^2$) | UAV | FP | 0.20 | TN | 7.57 |
| | GF-6 | | 0.33 | | 7.44 |
| | Landsat 8 | | 0.59 | | 7.18 |

4.1.1. Correctly Detected Landslide Areas (TP) and Correctly Detected Non-Landslide Areas (TN)

Of the 1.43 km$^2$ of landslides detected by the UAV images, the correctly detected landslide area was 1.23 km$^2$, accounting for 82.17% of the total area of landslides in the region; of the 1.11 km$^2$ of landslides detected by the GF-6 images, the correctly detected landslide area was 0.78 km$^2$, accounting for 70.42% of the total area of landslides in the region; and of the 1.32 km$^2$ of landslides detected by the Landsat 8 images, the correctly detected landslide area was 0.72 km$^2$, accounting for only 55.13% of the total landslide area in the region. In general, the area of correctly detected landslides decreases sharply as the image resolution decreases.

Correspondingly, the correctly detected non-landslide area was 7.77 km$^2$, the area of that by UAV images was 7.57 km$^2$, by GF-6 images it was 7.44 km$^2$, and by Landsat 8 images it was 7.18 km$^2$, respectively.

### 4.1.2. Misidentified Landslide Areas (FP)

Of all the landslide areas detected by the remote sensing images, some of them were non-landslide areas, among which the misidentified non-landslide area was 0.20 km$^2$ in the landslides detected by UAV images, 0.33 km$^2$ by GF-6 images, and 0.59 km$^2$ by Landsat 8 images. Because of UAV images' high resolution, most of the misidentified landslides were the boundaries around the landslides, and the area was small; the areas between the landslides were mostly misidentified in GF-6 images, while roads and other features were mostly misidentified as landslides in Landsat 8 images.

### 4.1.3. Undetected Landslide Areas (FN)

All types of images had landslide areas that were not correctly detected during landslide detection. Among them, the undetected landslide area by UAV images was 0.27 km$^2$, which was relatively small, while the undetected landslide areas by GF-6 images and Landsat 8 images were relatively larger, 0.71 km$^2$ and 0.77 km$^2$, respectively.

The undetected part in UAV images was mostly the edge of landslides with dense vegetation cover. Furthermore, individual large landslides were not completely broken after the occurrence, which left the vegetation on the landslide largely intact, so it was unable to correctly detect the complete landslide body by UAV images. In addition to the above-mentioned situation, the undetected areas that are common to GF-6 images and Landsat 8 images (Figure 5) were mostly distributed at the mouth of the gully and both sides of the river. These undetected areas' surface features were similar to the surrounding ravines and roads, and their landslide shape was mostly narrow and long, resembling the shape of the ravines. Due to the limitation of image spatial resolution, the mixed image elements in the medium and low-resolution images were widely present, causing the distinction between the features to be less obvious. This has made it difficult to detect landslides in these two types of images, because of the poor segmentation effect.

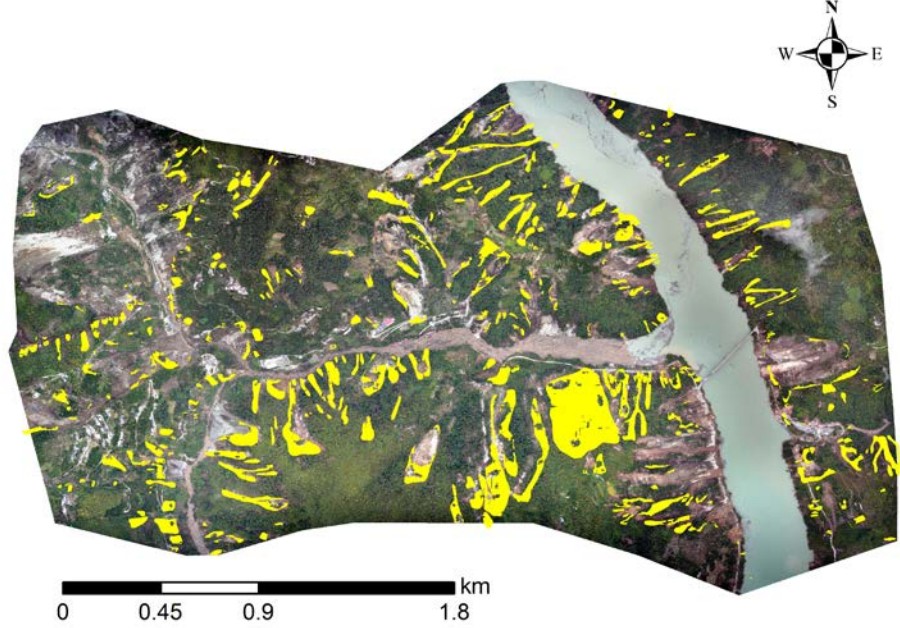

**Figure 5.** Undetected areas in GF-6 and Landsat 8 images.

*4.2. Detection Accuracy*

Based on the confusion matrix, five indicators were calculated and obtained (Table 2), including overall accuracy, positive accuracy, recall, F1 score, and average cross-merge ratio (mIoU), and were then compared for the detection accuracy of different types of remote sensing images.

Accuracy is the proportion of correct prediction results to the total results. The formula is as follows:

$$\text{Accuracy} = (TP + TN)/(TP + TN + FP + FN), \tag{7}$$

Precision is the ratio of the area of positive correct areas to the total positive areas. The formula is as follows:

$$\text{Precision} = TP/(TP + FP), \tag{8}$$

Recall is the ratio of correctly predicted landslide samples to all detected landslide samples. The formula is as follows:

$$\text{Recall} = TP/(TP + FN), \tag{9}$$

F1 Score is the harmonized average of precision rate and recall rate, and the formula is as follows:

$$\text{F1} = 2(\text{precision} \times \text{recall})/(\text{precision} + \text{recall}), \tag{10}$$

The average cross-merge ratio (mIoU) is the average of the ratio of the intersection and merge sets of the two sets of true and predicted values, and the formula is as follows:

$$\text{mIoU} = (TP/(TP + FP + FN) + TN/(TN + FP + FN))/2, \tag{11}$$

The statistics found that UAV images had a higher accuracy rate compared to both GF-6 images and Landsat 8 images. The overall accuracy rate characterizes the correctness of different images for landslide zone and non-landslide zone detection, including the true positive rate and true negative rate. The overall accuracy rate of UAV images reached 94.45%, while that of GF-6 and Landsat 8 images was similar at 88.78% and 85.22%, respectively. The positive accuracy of UAV images was the highest at 86.01%, while the positive accuracy of GF-6 images was reduced to 70.54%, and the positive accuracy of Landsat 8 images was 54.96%. Therefore, it can be seen that the spatial resolution of the image has a greater effect on the accuracy of landslide detection. The correct rate in landslide detection decreases sharply as the image resolution is reduced. Compared with GF-6 and Landsat 8 images, UAV images had a lower pseudo-positive rate and pseudo-negative rate for ETL detection, while the pseudo-positive rate in GF-6 and Landsat 8 images were similar at 7.66% and 8.31%, respectively. For the pseudo-negative rate, GF-6 images were lower than Landsat 8 images.

## 5. Comparison of Individual Detection of Landslide

The auto-detection only captured the area where landslides occurred but it did not distinguish different individual landslides. The area where ETLs developed would have a large number of landslides concentrated in patches. In order to study the effect of resolution on automatic landslide detection, this study used a manual method to distinguish different individual landslides from the automatically interpreted ones, especially by the criteria of geomorphological features to differentiate the landslides in mixed patches. The differences in automatically detected landslides based on remote sensing images of different resolutions were further analyzed for cataloguing the segmented landslides.

*5.1. Number and Area of Individual Landslides*

A total of 434 landslides were visually interpreted in the study area. Landslide areas were mainly concentrated in the range of 100–1000 m$^2$ and 1000–10,000 m$^2$, with 199 and 179 landslides, respectively, accounting for 87% of the total number of landslides. A total

of 15 landslides were smaller than 100 m$^2$ and 41 landslides were larger than 10,000 m$^2$. Among them, the smallest landslide was 29 m$^2$ and the largest landslide area was 84,191 m$^2$. The smallest landslide that could be deciphered by UAV images was 46.91 m$^2$, 73.54 m$^2$ by GF-6 images, and 105.14 m$^2$ by Landsat 8 images.

In general, the higher the resolution of the image, the greater the number of landslides detected. However, for the 199 landslides of 100–1000 m$^2$, UAV images detected 197 landslides, accounting for 98.99% of the landslides, while for GF-6 images and Landsat 8 images, the detection rate was improved, with 77 and 83 landslides, accounting for 39.07% and 41.71%, respectively. The details can be found in Table 3.

**Table 3.** Landslide detection results from remote sensing images.

|  | UAV | GF-6 | Landsat 8 |
|---|---|---|---|
| Total area of FCN detection/ m$^2$ | 1,425,870 | 1,110,561 | 1,322,209 |
| Accuracy | 94.45% | 88.78% | 85.22% |
| False positive rate | 2.91% | 7.66% | 8.31% |
| False negative rate | 2.16% | 3.56% | 6.36% |
| Precision | 86.01% | 70.54% | 54.96% |
| Recall | 82.00% | 52.67% | 48.32% |
| F1 Score | 83.96% | 60.30% | 51.42% |
| mIoU | 83.25% | 65.45% | 59.34% |

When detecting landslides, although landslides can be detected by remote sensing images of different resolutions, the detected areas of landslides can also vary. For example, the actual area of a landslide located at 102°9′6″ E, 29°31′52″ N was 21,680 m$^2$, while the area detected by UAV images was 20,762 m$^2$, 14,248 m$^2$ by GF-6 images, and only 10,975 m$^2$ by Landsat 8 images. The area proportion follows the area trend. For the landslides of 1000–10,000 m$^2$, the total area of landslides is 529,482.48 m$^2$, the UAV image can detect 80.16% of the landslides, and the GF-6 image and Landsat 8 image are still lower, 37.63% and 32.54%, respectively.

According to the above data, we found that the larger the landslide area was, the better the image detection would be in all three types of different-resolution images. The larger the landslide area is, the more obvious the change in the surface coverage is, so it is easier to be identified and interpreted by FCN. In order to analyze the effectiveness of landslide detection, the average detection rate was used, which is the average of all individual landslide detection rates within the given range.

In our study, Landsat 8 images could not detect landslides below 100 m$^2$. Only landslides between 100–1000 m$^2$ could be identified and interpreted, but only in small amounts. For landslides of 1000–10,000 m$^2$, the average detection rate of UAV images was 35.58% higher than that of GF-6 images and 36.86% higher than Landsat 8 images. For landslides larger than 10,000 m$^2$, the average detection rate of UAV images was 16.93% higher than GF-6 images and 21.81% higher than Landsat 8 images. (Figure 6)UAV images were more effective than the other two images in landslide detection regardless of area scales, and the smaller the area scale, the better the results would be. At the same time, the three types of images conformed to the understanding that the higher the image resolution, the better the detection. Only in detecting the area of 100–1000 m$^2$, the overall detection rate of GF-6 images was 17.11%, slightly lower than that of the Landsat 8 images, while the average detection rate is 9.08% lower due to some large-scale landslides in the Landsat 8 images that were not fully detected.

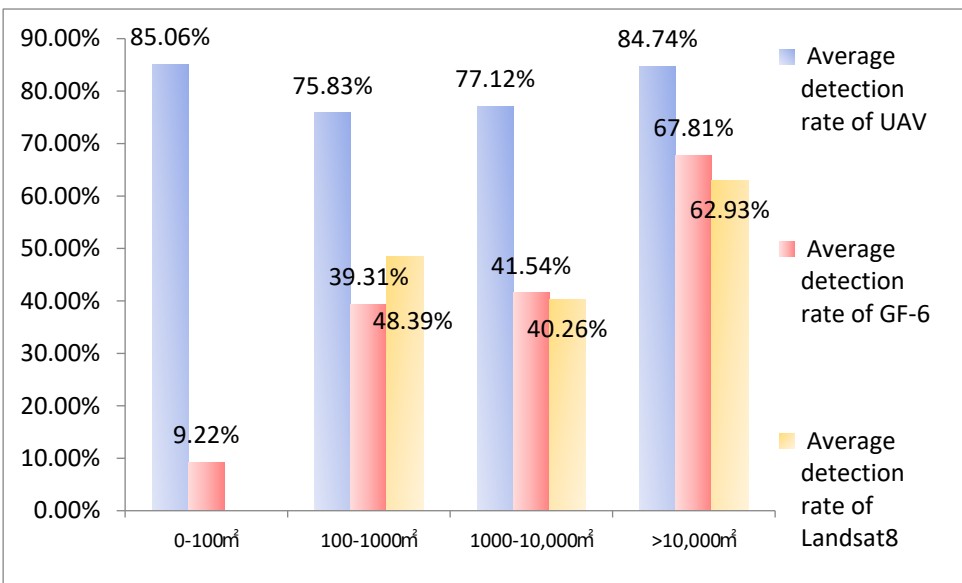

**Figure 6.** Average detection rate of each image.

### 5.2. Frequency–Area Curves

Landslide area–frequency curves can represent the proportional characteristics of landslides at different sizes and also reflect the overall magnitude of landslide events triggered by earthquakes [25,26]. The catalogued landslide data detected from UAV, GF-6, and Landsat 8 images were plotted for producing area–frequency curves to compare with the reference landslide data (Figure 7). The results showed that when a landslide area was larger than 2000 m$^2$, the area–frequency curves of the landslide detected by UAV, GF-6, and Landsat 8 images were more similar to those of the reference landslides with small differences.

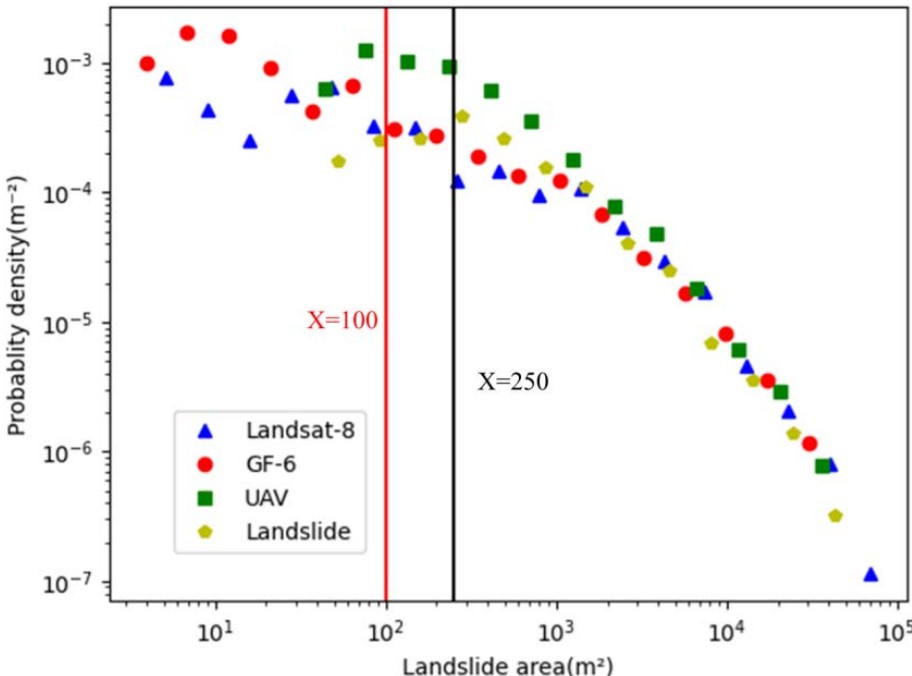

**Figure 7.** Frequency–area curves of landslides of remote sensing image.

When the landslide area was less than 2000 m$^2$, the area–frequency curves of landslides detected by different images showed larger differences. The inflection point of the area–frequency curve of the reference landslide data appeared at the position of 250 m$^2$, while for UAV images, the inflection point was located at the position of 100 m$^2$. Additionally, GF-6 and Landsat 8 images showed two inflection points. The landslide frequency from UAV image interpretation was higher than the reference curve in the interval between 250 m$^2$ and 2000 m$^2$ regarding the landslide area. The landslide frequency from GF-6 and Landsat 8 images was lower than that of the reference, it was mostly because some landslides in the area could not be detected, which thus reduced the frequency. Additionally, since some landslides could not be detected as complete, the broken and incomplete landslides were therefore detected and counted in a smaller range.

As for the landslides with an area of less than 100 m$^2$, there was a higher frequency of landslides detected by GF-6 and Landsat 8 images. According to the conclusion above, it was caused by incomplete detection of landslides in larger areas. Table 4 shows that GF-6 detected only one landslide within 100 m$^2$, while Landsat 8 images did not detect any landslides. Therefore, it suggests that the frequency of landslides in this part was composed of landslides that were incompletely detected in the range of 250 m$^2$ to 2000 m$^2$.

**Table 4.** Landslide indicators in each area of remote sensing images.

| | | | Landslide Area (m$^2$) | | | |
| | | | 0–100 | 100–1000 | 1000–10,000 | >10,000 |
|---|---|---|---|---|---|---|
| Landslide survey data | Visual detection field | Quantity | 15 | 199 | 179 | 41 |
| | | Area | 971.36 | 90,365.91 | 529,482.48 | 875,830.89 |
| | UAV | Quantity | 13 | 197 | 179 | 41 |
| | | Quantity proportion | 86.67% | 98.99% | 100% | 100% |
| | | Area | 644.90 | 67,136.95 | 424,454.14 | 737,596.27 |
| | | Area proportion | 66.39% | 74.29% | 80.16% | 84.21% |
| | Gf-6 | Quantity | 1 | 77 | 145 | 41 |
| | | Quantity proportion | 6.67% | 38.69% | 81.00% | 100% |
| | | Area | 6.78 | 15,464.37 | 199,229.51 | 567,406.37 |
| | | Area proportion | 0.70% | 17.11% | 37.63% | 64.78% |
| | Landsat 8 | Quantity | 0 | 83 | 141 | 41 |
| | | Quantity proportion | 0 | 41.71% | 78.77% | 100% |
| | | Area | 0 | 19,031.83 | 172,269.57 | 537,743.39 |
| | | Area proportion | 0 | 21.06% | 32.54% | 61.40% |

## 6. Error Analysis

### 6.1. Types of Automatic Remote Sensing Detection Errors

The types of errors that existed in landslide detection influenced by different resolutions were analyzed and summarized. Seven types of errors that existed in this study were found.(Figure 8) Considering the fact that landslide detection areas are hardly identical, the error in the interval of area 95%–105% is relatively small, and was therefore, not included in this paper for error statistics. Landslide detection areas that were smaller than the actual area had more errors in the three images. Especially for GF-6 and Landsat 8 images, due to the relatively low resolution of the images and the large image element size, the mixed image elements in the landslide boundary area had a greater influence on landslide detection. This error type existed in 77 places in UAV image-based landslide detection,

83 places in GF-6 images, and 97 places in Landsat 8 images, accounting for 35.00%, 23.31%, and 24.49% of the error quantity, respectively (Table 5).

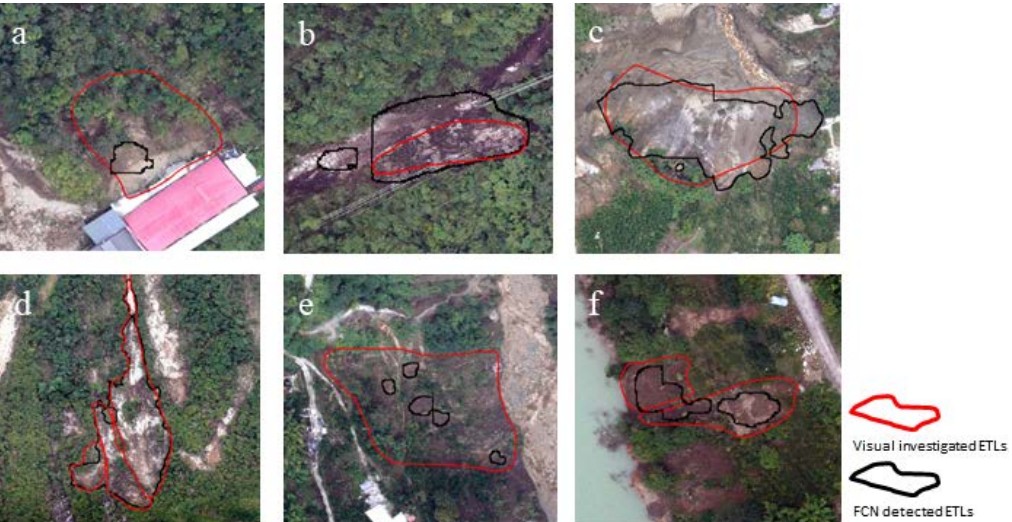

**Figure 8.** The type of automatic detection errors. (**a**) The landslide detection area in black line is smaller than the actual area in red. (**b**) The landslide detection area is larger than the actual area. (**c**) The detected area of the landslide partially coincides with the actual area. (**d**) A landslide is detected as multiple landslides. (**e**) Multiple landslides are detected as a single landslide. (**f**) Multiple landslides are detected as multiple landslides.

**Table 5.** Types of automatic detection errors in remote sensing images.

| Error Types | UAV | GF-6 | Landsat 8 |
|---|---|---|---|
| The single detected area of landslide is less than the actual landslide | 77 | 83 | 97 |
| The single detected area of landslide is more than the actual landslide | 17 | 19 | 35 |
| The single detected area of landslide partially coincides with the actual area | 29 | 19 | 24 |
| One single landslide is detected as multi | 13 | 7 | 2 |
| Multi landslides are detected as single | 67 | 53 | 58 |
| Multi landslides correspond to other multi landslides | 13 | 5 | 11 |
| Undetected landslides | 4 | 170 | 169 |

The appearance of the error of the landslide detection area larger than the actual area on UAV images was mainly a case of misjudging the gully at the foot of the landslide as a landslide, thus making the landslide detection area larger than the actual area. This type of error existed in 17 places in UAV image-based landslide detection, reaching 7.73% of its error quantity. Additionally, this type of error was found in GF-6 and Landsat 8 images, mainly because of the misjudgment of similar feature attributes, especially in areas where the landslide area was more similar to the surrounding environmental feature attributes, such as roads, open spaces, etc. This type of error was 19 and 35 on GF-6 and Landsat 8 images, reaching 5.34% and 8.84% of their error numbers, respectively. The number of landslide errors identified was highest for GF-6 and Landsat 8 images, with 170 and 169 errors, reaching 47.75% and 42.68% of their errors, while this error was less frequent in UAV images.

*6.2. Analysis of Spatial Distribution Errors*

The relative position of landslide identification errors was analyzed by comparing the unidentified landslide areas, the misidentified areas, and the mass center of the landslide areas.

The distance between the mass center of the undetected landslide area and that of the corresponding landslide was calculated (Figure 9). The origin position was the mass center of the landslide. The results showed that the area of the undetected landslides in UAV images was relatively small and mainly located in the area around the landslide far from the mass center of the landslide with an average distance of 17.61 m, and the farthest of 160.86 m. The area of the undetected areas in GF-6 and Landsat 8 images was relatively large and closer to the mass center of the landslide. When the undetected landslide area was located at the origin position, it means that the landslide was not detected as a whole. The average distance of the undetected area from the mass center of the corresponding landslide in GF-6 images was 11.76 m, and the farthest distance was 180.28 m. For Landsat 8 images, it was 13.65 m and 211.63 m, respectively.

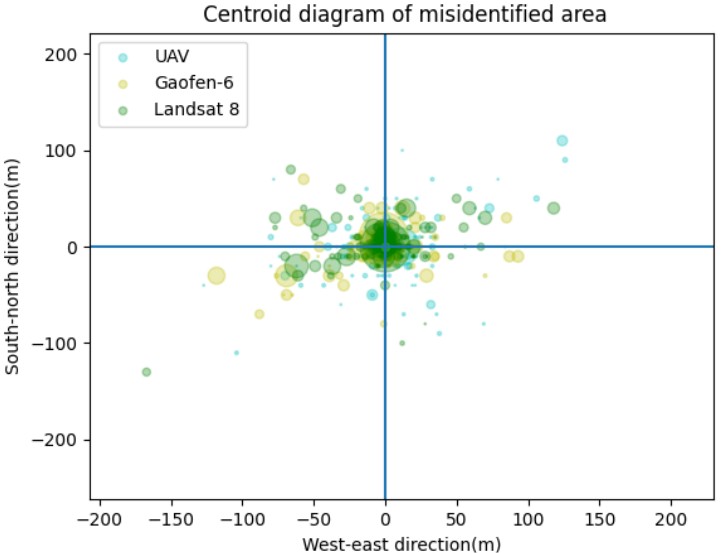

**Figure 9.** Schematic diagram of the location of each undetected area from the mass center of the landslide (The zero point position is the mass center of the complete landslide area. The positions of the points are relative to the mass center of the complete landslide area. The size of the circle represents the area of the undetected landslide).

The average distance between the misidentified area center and the corresponding landslide center (Figure 10) in UAV images was 35.07 m, with the nearest distance being 2 m and the farthest distance being 486.04 m; 36.45 m, 1.41 m, and 186.10 m for GF-6 images; and 103.14 m, 21.63 m, and 194.65 m for Landsat 8 images. UAV image-based data of each type were the largest among the three types of images, and the distribution was the most dispersed, which suggested that the misidentified parts in UAV images were those farthest from the boundary part of the landslide area.

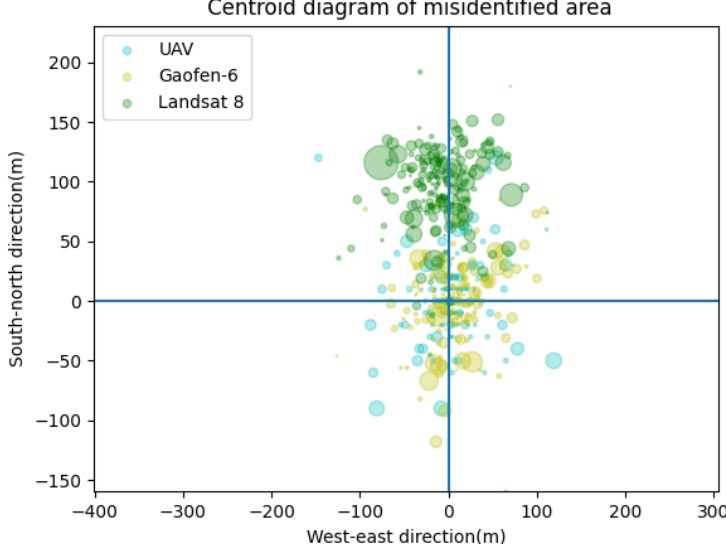

**Figure 10.** Schematic diagram of the location of each undetected area from the center of the landslide (The zero-point position is the mass center of the complete landslide area. The positions of the points are relative to the mass center of the complete landslide area. The size of the circle represents the area of the misidentified landslide).

### 6.3. Landslide Detection and Land Cover Type

The accuracy of automatic ETL detection is influenced by land cover types of the landslide occurrence area. The five main land cover types included in this study are forestland, high-coverage grassland, low-coverage grassland, built land, and water area (Table 6).

**Table 6.** Landslide rate under different surface coverage in remote sensing images.

|  |  | Forest | High-Coverage Grassland | Low-Coverage Grassland | Urban and Rural, Industrial and Mining, and Residential Land |
|---|---|---|---|---|---|
|  | Total area (m$^2$) | 1,205,622 | 2,587,019 | 4,155,431 | 366,290 |
| Visual detection | Landslide area (m$^2$) | 395,710 | 443,210 | 556,017 | 64,279 |
|  | Landslide area proportion | 27.12% | 30.37% | 38.10% | 4.41% |
|  | FCN-detected area (m$^2$) | 338,462 | 467,564 | 482,057 | 63,706 |
| UAV | Correctly detected area (m$^2$) | 294,644 | 402,229 | 427,352 | 56,940 |
|  | Accuracy | 87.05% | 86.03% | 88.65% | 89.38% |
|  | FCN-detected area (m$^2$) | 207,510 | 376,553 | 387,588 | 108,975 |
| GF-6 | Correctly detected area (m$^2$) | 175,506 | 258,710 | 282,867 | 46,147 |
|  | Accuracy | 84.58% | 68.70% | 72.98% | 42.35% |
|  | FCN-detected area (m$^2$) | 609,560 | 456,960 | 507,616 | 95,865 |
| Landsat 8 | Correctly detected area (m$^2$) | 140,753 | 243,160 | 282,400 | 44,909 |
|  | Accuracy | 23.09% | 39.89% | 46.33% | 7.37% |

In this paper, the number of landslides on each land cover type was counted. Some of the landslide areas were under various land cover types, and the area counts were performed for the areas within multiple different land cover types in this paper.

In the four feature types, the correct interpretation rate decreased as the resolution decreased. The effect of resolution on the interpretation results was particularly evident on the built land, for example, industrial and mining, and urban and rural residential areas. The accuracy of interpretation in UAV images in this feature type reached 89.38%, which was the highest among the four feature types; while for GF-6 images, it was only

42.35%, the lowest among the four feature types; and for Landsat 8 images, the accuracy rate decreased to 7.37%, which was much lower than the accuracy of interpretation in the other three feature types.

As the resolution decreased, the percentage of landslide interpretation increased under the feature types of high-coverage grassland. This is because when landslides occur in this feature type, the surface damage is more obvious and the landslide area changes significantly from the surrounding non-landslide areas, so it is easier to be interpreted. Under the feature type of high-coverage grassland, the interpretation accuracy in UAV images was 17.33% higher than that in GF-6 images and 46.14% higher than that in Landsat 8 images. The interpretation accuracy in GF-6 images was 28.81% higher than that in Landsat 8 images.

As the resolution decreased, the percentage of landslides interpreted under the forest feature type decreased continuously. The percentage of landslides interpreted under the forest feature type in Landsat 8 images was 9.35% lower than that in UAV images, and 1.95% lower than that in GF-6 images. (Figure 11) This is due to the fact that when landslides occur under this feature type, the surface damage is more obvious and the landslide area changes significantly from the surrounding area where no landslides occur, so it is easier to be detected. In contrast, for landslides under two feature types: low-coverage grassland and built land, Landsat 8 images detected 4.22% more landslides under the low-coverage grassland feature type than UAV images, and 3.34% more than GF-6 images; under the build land, Landsat 8 image-based detection results were higher than those of UAV images. Under the type of built land, Landsat 8 image-based detection result was 2.81% higher than that of UAV images and 1.58% higher than that of GF-6 images. The results suggested that as the resolution decreased, the proportion of landslide interpretation area with low vegetation coverage increased.

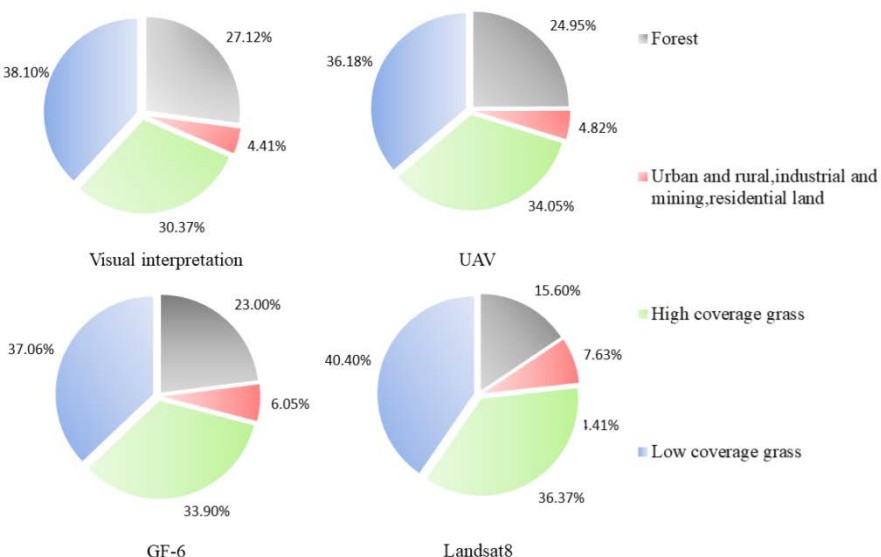

**Figure 11.** Proportion of landslides under different surface coverage in remote sensing images.

## 7. Conclusions and Discussions

We quantitatively compared and analyzed the ability and error types of remote sensing images with different resolutions: UAV images (resolution of 0.5 m), GF-6 images (resolution of 2 m), and Landsat 8 images (resolution of 15 m) in detecting the area and number of ETLs. The conclusions are as follows:

1. Differences in detecting ETL areas.

The resolution of the image plays a major role in automatic ETL detection. The higher the resolution of the image is, the higher the accuracy of landslide detection. In this study, the overall accuracy rates of UAV, GF-6, and Landsat 8 images were 94.45%, 88.78%, and

85.22%, respectively, and the precision rates were 86.01%, 70.54%, and 54.96%, respectively. The ETL area that was correctly detected in UAV images was 82.00%, against 52.67% and 48.32% when compared with GF-6 and Landsat 8 images, respectively.

2. Differences in detecting individual ETLs.

The ability of remote sensing images with different resolutions to detect individual ETLs varies greatly. The minimum landslide areas that were automatically detected in UAV, GF-6, and Landsat 8 images were 46.91 m$^2$, 73.54 m$^2$, and 105.14 m$^2$, respectively, in this study. The quantity of ETLs detected in UAV images was 99.07%, against 60.83% and 61.06%, compared with GF-6 and Landsat 8 images, respectively.

The detection rate of remote sensing images for landslides of different scales is quite different as well. In general, the larger the landslide area is, the higher the detection accuracy. The study showed that when the area of landslide was smaller than 100 m$^2$, 66.39% of the landslide was detected by UAV images, while for GF-6, it was only 0.70%. No landslide was detected by the Landsat 8 images. For landslide areas of 100–1000 m$^2$, 74.29% landslides were detected in UAV images, while for GF-6 and Landsat 8 images the rate was only 17.11% and 21.06%, respectively. For 1000–10,000 m$^2$, 80.16% of landslides were detected in UAV images, and only 37.63% and 32.54% for GF-6 and Landsat 8 images, respectively. For areas larger than 10,000 m$^2$, 84.21% landslides were detected by UAV images, and only 64.78% and 61.40% for GF-6 and Landsat 8 images.

3. Errors of detecting ETLs.

The main errors in automatic ETLs detection by remote sensing include six types: the detected area of a single landslide was smaller than the actual area, larger than the actual area, partially overlaps with the actual area, a single landslide was detected as multiple landslides, multiple landslides were detected as one single landslide, and multiple landslides corresponded to multiple landslides. These types accounted for different proportions in remote sensing images of different resolutions. For example, the main error in UAV images was that the detected area of a single landslide was smaller than the actual area, while for GF-6 and Landsat 8 images, the errors were mainly about undetected landslides. In addition, the positions of the undetected and incorrectly detected landslide areas relative to the actual landslide are also different. The higher the image resolution, the closer the undetected landslide area is to the actual landslide, while the incorrectly detected landslide area is to the actual landslide.

The land cover has a great influence on landslide detection. ETLs are mainly distributed in low-coverage grassland, high-coverage grassland, and forest areas. In this study, low-coverage grassland, high-coverage grassland, and forest areas accounted for 38.10%, 30.37%, and 27.12% of the total landslide area, respectively, while landslides in built land occurred less. The GF-6 image had a higher detection accuracy of landslides in forestland, followed by low-coverage grassland and high-coverage grassland, and a lower detection accuracy of landslides in built areas; Landsat 8 images had a relatively high detection rate of landslides in low-grass covered areas, followed by grassland and forestland, but the detection rate was relatively low. The most prominent is that the detection of landslides on built land was particularly low, which also shows that Landsat 8 images cannot distinguish well between built land and landslides. Compared with GF-6 and Landsat 8 images, the image of land cover types detected in UAV images was less affected by the land cover type, including landslides on built land, which also had a high detection rate.

When carrying out ETLs surveys, the number and area of actual seismic landslides can be inferred using the results of this study based on the number of landslides interpreted from low- and medium-resolution remote sensing images for some areas where aerial photography is not available. In addition, this method is mainly based on the detection of land cover, so it is also applicable to rainfall-type landslides after rapid sliding, but for slowly deforming landslides, the detection method in this paper is not applicable because of their insignificant surface changes.

Obtaining a complete ETLs inventory by automatic detection is more difficult than visual detection. At present, many factors affect the accuracy of landslide detection [27,28].

For example, the FCN detection method takes pixels as the basic unit, thus producing a large number of debris polygons. It is still time-consuming and laborious to integrate and eliminate them manually. The detection accuracy of ETLs from medium-resolution images still needs to be improved. Further improvements to the algorithm model are needed. This study illustrated the ability and accuracy of ETL detection by remote sensing images of different resolutions on the one hand, and statistically analyzed the types of errors in ETL detection by remote sensing images with different resolutions. On the other hand, the model can be further optimized in order to provide the detection accuracy of ETLs by remote sensing. Additionally, the algorithm for debris polygons generated by landslide edges can be improved by adding some edge detection algorithms to make landslide identification more accurate.

**Author Contributions:** Conceptualization, J.Z. and Y.H.; methodology, Y.H.; software, Z.M.; validation, L.Z.; formal analysis, Y.H.; investigation, L.Z.; resources, J.Z.; data curation, Z.M.; writing—original draft preparation, Y.H.; writing—review and editing, J.Z., L.Z. and R.L.; visualization, L.Z.; supervision, J.Z., H.H., R.C. and Y.G.; project administration, J.Z.; funding acquisition, J.Z. All authors have read and agreed to the published version of the manuscript.

**Funding:** This research was supported by the "Second Tibetan Plateau Scientific Expedition and Research Program (STEP)" (Grant No. 2019QZKK0902) and the Strategic Priority Research Program of the CAS (No. XDA23090203).

**Data Availability Statement:** Not applicable.

**Conflicts of Interest:** The authors declare no conflict of interest.

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
