# Peer review of "How Spatial Resolution of Remote Sensing Image Affects Earthquake Triggered Landslide Detection: An Example from 2022 Luding Earthquake, Sichuan, China"

_land, doi:10.3390/land12030681_

Round 1

Reviewer 1 Report

The manuscript is written on a relevant topic. Scientific studies demonstrate the potential of different resolution remote sensing data for landslide detection, and highlight the importance of selecting the appropriate resolution data. However, before the article is published, significant edits are needed.

Main remarks are to the description of the work methodology; therefore, it is difficult to evaluate the correctness of the results.

Title  - hyphenation in the title is not allowed. The title of the article should indicate the study area location. Since the introduction deals with the problem of ETL recognition, it is worth reflecting this in the title

For example:

How spatial resolution of remote sensing image affects earthquake-triggered landslide detection (2022 Luding landslide, Sichuan, China)

Abstract and Introduction

Line 22-23 - No  study  has  yet  been  done  to  give  a  quantitative  assessment  of  such  variation and its impact on landslide detection. – Unsubstantiated statement!!! The use of remote sensing data and DEM to identify landslides has been used for a long time. If you prove the opposite, then you need to substantiate this statement in the Introduction section, giving examples of sharing data of different resolutions, indicating their advantages and disadvantages.

Here are some examples of case studies that demonstrate the use of different resolution remote sensing data and DEM for landslide detection:

https://iopscience.iop.org/article/10.1088/1755-1315/169/1/012083

https://shs.hal.science/halshs-00467545/document

https://www.sciencedirect.com/science/article/pii/S0013795208001786?via%3Dihub

https://www.tandfonline.com/doi/abs/10.1080/01431160512331314047?journalCode=tres20

2. Study area

Specify the number of landslide events used in the work

Fig.1 Add a coordinate grid. Need to unify legend and scale bar. All cartographic material needs to have coordinate grid, scale bar, orientation, legend.  a). it's not a topographical, but hypsometric map. Legend, scales and inscriptions for maps d. e. unreadable. It is desirable to remake this figure, since it does not correctly show the location and boundaries of the study area relative to the borders of the country and region, as well as the coverage of remote sensing data.

2.2 Multi remote sensing image – strange formulation. Change to Remote sensing data. Or Multisource remote sensing data

Line 32 - For Landsat 8 the resolution of 15m is typical for panchromatic images, but you obviously used a multispectral image.

Table 1 - UAV –  Is this a photo or an orthophotomap? Are there many of them or just one?

Fig.2 - As can be seen from the landslide boundary in Fig. a-b referencing accuracy of UAV and satellite images is different

3. Methods 

3.1 Deep fully convolutional neural networks

What software was used for the FCN procedure?

3.2 Adam (Adaptive Moment Estimation) optimizer - the section describes the optimizer, the standard procedure, the detailed description does not make sense.

Line 200-204 - pixel point - looks like that tautology. Considering the different resolution of images obtained by different sensors, it is necessary either to unify the resolution or use a different segment size. 500x500 - why was this segment size chosen? Otherwise, with the same segmentation, the sample size for different images will be different. It is necessary to describe in detail the procedure for preparing and processing data, indicating the size of the sample.

In what software was the segmentation carried out; was the intersection between the segments established; in what format was the training sample formed - raster + raster or raster + text?

It seems that the training set and the test set are mixed up. Typically, the training sample is 70%, and the test sample is 30%.

Fig. 6 Labels for the scales of the diagram are required

Fig 7 – UAV instead of AUV

Please do not end the chapter with the picture.

Fig 8. Provide graphical legend – landslide area, Actual and Detection

In Сonclusion, it is desirable to draw a conclusion about which of the remote sensing data used are most applicable for determining landslides, taking into account the spatial resolution, coverage, cost and efficiency of surveying.

Reviewer 2 Report

1. In the abstract section, lines 22-23, the author stated that “No study has yet been done to give a quantitative assessment of such variation and its impact on landslide detection.”. This statement is inaccurate; please summarize the reference and give an appropriate conclusion.  

2. Figs 1a and 1b were not well plotted. Please try to make the figures clearer. In Fig.1a, the legend is unclear and the words are even overlap; in Fig.1b, the map cannot clearly show the studied area. In addition, Figs 1d and 1e need some modifications to make them clearer.  

3. The literature review is incomplete. Though the author stated that “And such differences in landslide detection from multi-source remote sensing images are rarely addressed.”, the author should summarize the application scenarios of remote sensing images with different types and resolutions in dealing with ETLs. Besides, before giving detailed examples, it’s important to first summarize the main investigating methods for automatic interpretation of ETLs. In addition, the machine learning methods applied in landslides were missed and should be addressed. 

4. Lines 62 to 64 are confusing. Please clarify.

5. This paper is lack of a discussion on the knowledge already available in the literature. Besides, the significance of the achieved results should also be addressed.

6. To study the differences between images of different resolutions for ETL detection, three groups of images with different resolutions were analyzed. What about the applicability of the achieved results in other types of landslides? For examples, the rainfall induced and freeze-thaw induced landslides. Could we refer to this paper in the detection of rainfall induced and freeze-thaw induced landslides?

7. The differences in the detection of landslides were ascribed to different resolution by the three means. This reason seems common sense. Besides this reason, are there any other factors affecting landslide detection?

8. How to remedy the detection errors caused by the three methods? This point should also be addressed in the paper. 

Round 2

Reviewer 1 Report

The authors have substantially corrected the manuscript, but minor revisions are needed.

Figure.1 - Low resolution drawing, which makes it difficult to read the inscriptions. Figure captions (a b c d) are not evenly distributed. It is necessary to redo the drawing according to the logic of perception - c) and d) move to the left side and rename to a) and b). Accordingly a) and b) move to the right side and rename to c) and d). The scales in figures a) and b) are of different sizes with the same degree grid. Different sized graticule ticks on a) and b). Why is the maximum value of the scale 1.2 km and not 1 km? Since a) and b) show the same area indicated in image d), the boundaries of the graticule must be the same. Moreover, the graticule is shifted in one of the figures, see the lines in the image. Please provide the Figure, that show the coverage of remote sensing and UAV images.

Table 1 – Change type for UAV – orthophotos, provide number of them Line 218. ‘All workflows are implemented in python.’ since Python™ is the name of the program, it should be capitalized. Add Trade Mark and software version Line 211- I draw your attention again - 'pixel point' - looks like that tautology. Delete the 'point'. Figure 11 – Change 'industrial and mining, and urban and rural residential' on 'built land'

In conclusion, I would like to wish good luck and speedy publication of the article!
